# Synergistic Hypolipidemic Effects and Mechanisms of Phytochemicals: A Review

**DOI:** 10.3390/foods11182774

**Published:** 2022-09-09

**Authors:** Yazhou Liu, Chunlong Liu, Xiaohong Kou, Yumeng Wang, Yue Yu, Ni Zhen, Jingyu Jiang, Puba Zhaxi, Zhaohui Xue

**Affiliations:** 1School of Chemical Engineering and Technology, Tianjin University, Tianjin 300072, China; 2Food and Drug Inspection and Research Institute of Tibet Autonomous Region, Lhasa 850000, China; 3Dynamiker Biotechnology (Tianjin) Co., Ltd., Tianjin 300450, China

**Keywords:** synergistic hypolipidemic effect, phytochemicals, action mechanism, structure-bioactive relationship, combination, lipid metabolic pathways

## Abstract

Hyperlipidemia, a chronic disorder of abnormal lipid metabolism, can induce obesity, diabetes, and cardiovascular and cerebrovascular diseases such as coronary heart disease, atherosclerosis, and hypertension. Increasing evidence indicates that phytochemicals may serve as a promising strategy for the prevention and management of hyperlipidemia and its complications. At the same time, the concept of synergistic hypolipidemic and its application in the food industry is rapidly increasing as a practical approach to preserve and improve the health-promoting effects of functional ingredients. The current review focuses on the effects of single phytochemicals on hyperlipidemia and its mechanisms. Due to the complexity of the lipid metabolism regulatory network, the synergistic regulation of different metabolic pathways or targets may be more effective than single pathways or targets in the treatment of hyperlipidemia. This review summarizes for the first time the synergistic hypolipidemic effects of different combinations of phytochemicals such as combinations of the same category of phytochemicals and combinations of different categories of phytochemicals. In addition, based on the different metabolic pathways or targets involved in synergistic effects, the possible mechanisms of synergistic hypolipidemic effects of the phytochemical combination are illustrated in this review. Hence, this review provides clues to boost more phytochemical synergistic hypolipidemic research and provides a theoretical basis for the development of phytochemicals with synergistic effects on hyperlipidemia and its complications.

## 1. Introduction

Hyperlipidemia is a disorder in which abnormal lipid metabolism results in a higher-than-normal level of one or more lipids in the serum, and the common symptoms are high levels of total serum cholesterol (TC), triglycerides (TG), and low-density lipoprotein cholesterol (LDL-C) or low levels of high-density lipoprotein cholesterol (HDL-C), which is called dyslipidemia in modern medicine [1,2,3]. Hyperlipidemia can cause some serious cardiovascular diseases, such as coronary heart disease (CHD) and atherosclerotic cardiovascular disease (ASCVD), which are responsible for millions of deaths in the world every year [4]. Alarmingly, a study of early subclinical atherosclerosis showed that 63% of participants have symptoms of subclinical atherosclerosis [5]. Although considerable progress has been made in the treatment of hyperlipidemia, the incidence rate and risk associated with this disease are still rising. Therefore, the prevention and treatment of hyperlipidemia are extremely important. Currently, the main treatment of hyperlipidemia is chemical drugs, and the classical lipid-lowering drugs include statins, fibrates, and nicotinic acids [6]. The above-mentioned chemical drugs have definite clinical effects and obvious effects in lowering blood lipid, but similar to other chemical drugs, long-term use of statins may also cause a series of potential side effects, for example, they may lead to liver and kidney function damage, gastrointestinal reactions and other adverse reactions [7,8,9]. In addition, the regulation of lipid metabolism is a complex process involving multiple pathways and targets, and it is difficult for the current single-target lipid-lowering drugs to achieve both primary and secondary effects [10]. Therefore, safe and effective substitutes are urgently needed to treat hyperlipidemia and its related complications. In this context, phytochemicals have received considerable attention for their safety and therapeutic potential [11,12].

Studies have shown that the natural active ingredients in some plants have unique advantages in treating hyperlipidemia and preventing the development of cardiovascular disease [13]. Furthermore, many studies have emphasized the hypolipidemic benefits of phytochemicals, which are multi-component, multi-targeted, and have relatively low toxic effects [14]. In fact, plants are natural sources of medicines, and their roots, stems, leaves and seeds are rich in polysaccharides, flavonoids, saponins, phytosterols, fatty acids, phenols, polypeptides and other small molecular compounds, which are active ingredients in drugs for cardiovascular diseases [15,16]. Many in vitro and animal studies have shown that the consumption of these bioactive phytochemicals significantly improves hypertension, low-density lipoprotein oxidation, lipid peroxidation, total plasma antioxidant capacity and dyslipidemia [17]. After years of verification of hypolipidemic phytochemicals, the structural properties of their main components are becoming clearer, and the mechanism of action is becoming clearer. For example, it has been suggested that plants contain many biologically active phytochemicals that can act on multiple targets in complex disease networks. In addition, different phytochemicals act synergistically at each target to intervene in the development of disease and ultimately achieve therapeutic effects [18,19]. Interestingly, it is not clear which specific components produce the actual effects, which makes the study of synergistic effects among phytochemicals a hot topic of interest.

Currently, most studies on phytochemicals to lower blood lipids are generally limited to single substances or compounds of the same categories, while a single food may contain several or even hundreds of active phytochemicals [20]. Food intake is also generally several substances in the body at the same time. Therefore, the study of the synergistic effects of these active phytochemicals has important practical significance and is also a new area of research in nutrition and functional foods. However, due to the variety and structural complexity of phytochemicals in foods, many factors influence their digestion, absorption and utilization in vivo [21]. Although research relevant to interactive effects among the phytochemicals has mounted up, the mechanism of phytochemicals synergy is still not clear. Especially, biological influence factors are often neglected [22]. Many researchers believe that one way to solve these problems is to combine two or more phytochemicals to see whether and how they work together to lower blood lipids [23,24]. 

Lipid metabolism is a complex process that involves lipid biosynthesis, absorption, transport, and elimination [14]. Due to the complexity of the lipid metabolism regulatory network, the synergistic regulation of different metabolic pathways or targets may be more effective than a single pathway or target in the treatment of hyperlipidemia. In recent years, the process of lipid metabolism has become more and more clear, and some important pathways and targets have been discovered by researchers. For example, the 5‘-adenosine monophosphate (AMP)-activated protein kinase is a natural energy sensor in mammalian cells that plays a key role in lipid metabolism [25]. This lays the foundation for us to study the synergistic lipid-lowering effects of phytochemicals.

Recent research has shown that synergistic effects are the basis of the action of most phytochemicals in biological systems. The complex interactions of phytochemicals facilitate the combined action on multiple cellular and molecular targets, leading to measurable biological effects. These effects can be observed among the components of plant extracts and can be used to develop safer and more effective treatment strategies to treat and prevent diseases [26]. For example, Park et al. showed that treatment with the combination of *Hippophae rhamnoides* and *Zingiber mioga* extracts inhibited the expression of genes associated with adipocyte differentiation in 3T3-L1 cells more than any single extract treatment, and the combination inhibited lipid accumulation in hepatocytes more effectively than the single extract. However, which phytochemicals of *Hippophae rhamnoides* and *Zingiber mioga* have synergistic effects needs to be further investigated [27]. Similarly, Yari et al. investigated the clinical efficacy of flaxseed and hesperidin in non-alcoholic fatty liver disease and observed significant synergistic effects of the combination on the regulation of lipid metabolism [28]. 

The concept of phytochemical synergy plays an important role in promoting human health and preventing hyperlipidemia. In this review, we summarize for the first time the synergistic hypolipidemic effects of different phytochemical combinations and the possible mechanisms of the synergistic hypolipidemic effects of phytochemical combinations, which provide clues for further development of phytochemical functional foods with synergistic effects and hypolipidemic potential.

## 2. Synergistic Hypolipidemic Effects between Phytochemicals of the Same Category

Increasing evidence suggests that combinations of phytochemicals from the same category may have a stronger hypolipidemic effect than a single phytochemical. Table 1 lists some of the combinations of the same category of phytochemicals that are now attracting scientific attention, identifying the categories with the potential hypolipidemic effects.

### 2.1. Synergistic Hypolipidemic Effects of Flavonoids

Flavonoids are a very rich and diverse category of natural phytochemicals with important biological activities, which are composed of a common diphenylpropane (C6-C3-C6) skeleton in which two aromatic rings are linked by a three-carbon chain [38]. Most flavonoids can be sub-classified into the following categories, namely flavones, flavonols, flavanones, flavanols, isoflavones, and anthocyanins (Figure 1) [39]. Flavonoids are widely distributed in our daily diet and are the major phytochemicals in foods such as vegetables, fruits, tea, and cocoa [40], and they can exist as free aglycones but are usually combined with glycosides and dissolve in water in this form [41]. Because of the potential health care value, safety and medicinal significance, it is considered to be an indispensable ingredient in all kinds of medicines and dietary supplements [42]. 

Studies have shown that eating foods rich in flavonoids is associated with lower cardiovascular risk because of a significant reduction in cholesterol levels and free radical scavenging activity [43,44]. Flavonoids can regulate the imbalance of lipid metabolism by inhibiting lipid peroxidation and endogenous lipid biosynthesis and promoting lipid redistribution and exogenous lipid metabolism, significantly reducing TG, TC and LDL-C levels [45]. For example, to investigate the interrelationship among flavonoids in the antioxidant and hypolipidemic effects, Qin et al. used ever-red and ever-green leaves during the development of crabapple cultivars. They identified a total of 16 flavonoids from them and predicted a positive interaction of flavonoids in ever-red by principal component analysis, and the experimental results also showed that the total antioxidant capacity was significantly higher than the sum of the antioxidant capacity of individual compounds [46].

Quercetin and kaempferol (different flavonoids), found in high levels in fruits and vegetables, have been shown to protect against cardiovascular diseases by regulating lipid levels [47]. Yusof et al. evaluated the lipid-lowering potential of quercetin and kaempferol by LDL-C uptake on HepG2 cells. They found that the mixture of quercetin and kaempferol (1:1,2:1 and 1:2) decreased the cell viability more than treatment individually, and the combination of quercetin and kaempferol in a ratio of 1:1 had the best effect on the LDL-C uptake of HepG2 cells. The conclusion just shows that quercetin and kaempferol have some synergistic effects [29].

In addition, Ma et al. isolated 12 flavonoids (jaceosidin, kaempferol, chrysoeriol, quercetin, apigenin, hispidulin, luteolin, quercitrin, rutin, isorhamnetin, genkwanin, and acacetin) from *Artemisia sacrorum*, which were arranged into 11 combinations to investigate their synergistic inhibitory effects on lipid accumulation in 3T3-L1 cells, respectively. Combined analysis of oil-red O staining, triglyceride levels, and lipid accumulation assays showed that the combination of acacetin and apigenin (1:1) had a more significant synergistic inhibitory effect on lipid accumulation compared to the compounds used alone [30]. In addition, the combinations of the plant flavonoids rutin and epicatechin (1:3) were tested on alloxan-induced diabetic mice for 28 days. The combination showed impressive anti-diabetic, anti-oxidant and anti-inflammatory activity without any observed signs of toxicity, and the formulation is considered to be a potentially safe, multi-target drug alternative [31]. 

Sea buckthorn is rich in flavonoids, which have hypolipidemic and hypoglycemic effects in mice fed with a high-fat diet [48]. To investigate the ameliorative effect of sea buckthorn flavonoids on obesity and hyperlipidemia, a network of component-target-disease was constructed by screening 12 biologically active flavonoids and 60 target sites using network pharmacological analysis and in vitro experimental methods. It has been shown that four bioactive flavonoids, including quercetin, catechin, hesperidin and isorhamnetin, may synergistically improve hyperlipidemia by promoting the conversion of cholesterol to bile acids and cholesterol efflux, inhibiting the de novo synthesis of cholesterol, and accelerating fatty acid oxidation [32]. 

The combination of plant flavonoids may have diverse biological activities such as antioxidant, antibacterial, hypolipidemic, immune regulation, and liver protection [49]; we can obtain a new, safe and multi-objective combination of plant flavonoids by optimizing the combination and proportion of them, and the synergy of these phytochemicals will have broad application prospects in the fields of medicine and plant-based functional foods.

The basic structures of reported flavonoids with synergistic hypolipidemic effects from plants are shown in Figure 1.

### 2.2. Synergistic Hypolipidemic Effects of Polysaccharides

Polysaccharides are carbohydrates composed of more than 10 monosaccharides linked by glycosidic bonds [50] and are one of the important active components of plants. They have the activities of hypolipidemic, hypoglycemic, enhancing immunity, anti-oxidation, anti-inflammation, anti-atherosclerosis and so on [51,52]. In general, polysaccharides can be divided into two categories: homo-polysaccharides and hetero-polysaccharides. A typical homo-polysaccharide is defined as having only one monosaccharide repeating on the chain, while a hetero-polysaccharide is composed of two or more categories of monosaccharides (Figure 2) [53]. Polysaccharides have the characteristics of strong polarity, large molecular weight and difficulty to confirm the structure [54]. It has been suggested that polysaccharides can inhibit the absorption of exogenous lipids and accelerate the hepatic catabolism of TC by physically binding to lipid molecules or bile salts in the gastrointestinal tract. The larger the relative molecular mass, the greater the characteristic viscosity or hydrophobicity of polysaccharides, and the stronger the corresponding binding ability [55]. Zhang et al. [56] claimed that some polysaccharide compounds can bind to each other to enhance the affinity between the receptor and the polysaccharide or to activate more polysaccharide binding sites on the receptor, thus resulting in a synergistic effect.

To investigate the physicochemical properties, anti-inflammatory and hypolipidemic effects of different polysaccharides in lipopolysaccharide-induced RAW264.7 macrophages, a multifactorial test was conducted using three different concentrations (0, 50 and 100 μg/mL) of high molecular weight dextran (885.2 kDa) and low molecular weight heteropolysaccharide (24.5 kDa). The results showed that high molecular weight dextran and low molecular weight heteropolysaccharide alone did not have significant effects compared to the control group, while the combination showed significant inhibitory effects, indicating a significant synergistic effect between them [33]. Similarly, Deng et al. analyzed the effect of complex polysaccharides and their combinations on RAW 246.7 macrophages, showing that complex polysaccharides with molecular weights between 100 and 1000 kDa had higher activity compared to the corresponding single-component polysaccharides, which also suggests a synergistic effect of different polysaccharides [57]. 

In addition, Li et al. investigated the effects of dietary fiber from bamboo shoots on hyperlipidemia mice induced by a high-fat diet. After 6 weeks of treatment with the combination of soluble dietary fiber (SDF) and insoluble dietary fiber (IDF), the body weight, body fat and adipose tissue mass of rats were significantly reduced (*p* < 0.05), and TC, TG and LDL-C levels were reduced by 30.20%, 53.28%, and 35.63%, respectively, compared with the model group. SDF + IDF (1:1) treatments had a better ability of lowering blood lipid and showed synergistic effects in preventing hyperlipidemia [34]. This synergistic effect of the combination may be related to their vast array and saccharide-based complex structures [58].

The studies of the active mechanism and the structure–activity relationship are the basis of the application of polysaccharides. However, compared with other biomolecules, polysaccharide structures are more complex, resulting in many polysaccharide structures with significant activity that cannot be easily identified. Moreover, the absorption, transport, distribution and metabolic processes of polysaccharides in vivo are difficult to investigate, which greatly limits the development of polysaccharides in the direction of hypolipidemia. At present, the hypolipidemic activity is mainly aimed at plant crude polysaccharides, containing a mixture of polysaccharides, oligosaccharides, polysaccharide proteins and other components, which makes researchers face certain difficulties in the study of their synergistic effects. In recent years, with the development of new science and technology, the derivatization modification of natural polysaccharides or the artificial synthesis of polysaccharides with well-defined structures are expected to be used to investigate the structure–activity relationship and activity mechanisms of polysaccharides [59]. Therefore, in the future, strengthening the research on the structure, active groups and structure–activity relationship of polysaccharide molecules will be the key to elucidating the synergistic effects of different plant-derived polysaccharides.

### 2.3. Synergistic Hypolipidemic Effects of Polyphenols

Polyphenols are secondary metabolites produced by many edible plants and have anti-diabetic, anti-inflammatory, anti-oxidant and hypolipidemic capabilities [60]. As an anti-oxidant, polyphenols are able to reduce oxidative damage to lipids, proteins, enzymes, carbohydrates and DNA in living cells and tissues, which is mainly attributed to the ability to scavenge free radicals, provide hydrogen atoms or chelate metal ions [61]. The combination of several polyphenol components could improve antioxidant and hypolipidemic efficiency, which can expand their applications in nutrition and biomedicine. For example, Heo et al. found that individual phenolic compounds showed their specific antioxidant capacity, and the sum of the antioxidant capacity of phenolic compounds resulted in an increase in total antioxidant capacity [62].

Proanthocyanidins and pterostilbene are natural phenolic antioxidants with hypolipidemic effects [63,64,65]. Hannan et al. studied their hypolipidemic effects combined with nicotinic acid in cholesterol-fed rabbits. The results showed that the LDL/HDL ratio and atherogenic index were suppressed significantly in blend therapies with maximum effects of 59.3 and 25% (*p* > 0.001) observed in 50:30:20 ratios of OPC, NA and PT compared to individual therapies 37 and 18% max respectively [35]. This study provides important evidence for the synergistic advantage of polyphenols in the hyperlipidemia effect and its complications. In addition, four phenolic compounds, including catechin, hesperidin, ferulic acid and quercetin, were also exhibiting synergistic effects in the prevention of low-density lipoprotein oxidation in humans [36].

At present, there are few studies on the synergistic effects of polyphenols on lowering blood lipid, which is mainly because there are many hydroxyl groups in the structure of polyphenols, which makes it very unstable in light, heat, and alkaline conditions [66]. Furthermore, many polyphenols are poorly soluble and have low bioavailability in humans [67], which limits their commercial use in functional foods.

The basic structures of reported polyphenols with synergistic hypolipidemic effects from plants are shown in Figure 3.

### 2.4. Synergistic Hypolipidemic Effects of Other Phytochemicals

In addition to the above synergistic hypolipidemic effects among the same phytochemicals category, other phytochemicals such as amides have also been found to have synergistic effects in the treatment of hyperlipidemia and related diseases. Chen et al. investigated the synergistic effects of different mass ratios of numb-tasting components of *Zanthoxylum bungeanum* and capsaicin on lipid levels in hyperlipidemic mice. Compared with the control group, feeding three different mass ratios (1:8, 2:7, and 3:6) of numb-tasting components of *Zanthoxylum bungeanum* and capsaicin reduced the serum levels of TC, TG, and LDL-C in mice (*p* < 0.05) and the symptoms of fatty liver in rats. Among them, the best effect was achieved at 3:6, without affecting the normal development of the mouse liver [37]. 

The basic structures of reported other phytochemicals with synergistic hypolipidemic effects from plants are shown in Appendix A.

In summary, a certain amount of studies have reported on the synergistic hypolipidemic effects of the same category of phytochemicals, especially flavonoids and polysaccharides. Nevertheless, there are still some unresolved aspects; for example, the phytochemical synergistic effects are usually studied based on observations in animal models, while in-depth and systematic analyses at the molecular level are still pending. In addition, we can search for new combinations of phytochemicals with clear molecular structures and active groups showing synergistic hypolipidemic effects, which should serve as the focus of further investigations.

## 3. Synergistic Hypolipidemic Effects between Different Categories of Phytochemicals

Nowadays, a large number of researchers indicated that different categories of phytochemicals could show synergistic hypolipidemic effects. Table 2 lists recent studies based on the hypolipidemic interactions between different categories of phytochemicals.

### 3.1. Synergistic Hypolipidemic Effects of Flavonoids with Other Categories of Phytochemicals

Quercetin (3,3′,4′,5,7-pentahydroxyflavone) is a flavonol compound with a wide distribution in the plant kingdom that has a variety of biological activities [81]. Resveratrol (3,5,4′-trihydroxytrans-stilbene) is considered as a natural antioxidant and is known for its anti-atherosclerotic properties, inhibiting lipid peroxidation and enhancing cholesterol efflux [82]. Arias et al. investigated the additive or synergistic effects of resveratrol and quercetin on fat accumulation and triglyceride metabolism in mice fed a high-fat diet. Mice were treated with resveratrol (15 mg/kg/d), quercetin (30 mg/kg/d), or a combination of them for 6 weeks, respectively. The results showed that the combination with quercetin or resveratrol resulted in a significant reduction in lipid accumulation compared to treatment alone, and the reduction percentage was greater than the calculated additive effect [83]. 

Similarly, Yang et al. observed that in maturing preadipocytes, resveratrol and quercetin individually suppressed intracellular lipid accumulation by 9.4% and 15.9%, respectively, and the combination of them at the same dose decreased lipid accumulation by 68.6% [84]. Furthermore, a gas chromatography-mass spectrometry (GC-MS)-based metabolomic approach was used to assess the potential role and mechanisms of quercetin and resveratrol combination (2:1) at different doses (45, 90 and 180 mg/kg) in high-fat diet (HFD)-induced obese rats. A total of 22 differential metabolites were found at the transcriptional and metabolic levels in the HFD group compared to the normal group, involving amino acid, galactose and pyruvate metabolism, pantothenic acid and coenzyme a biosynthesis, citric acid cycle, and lysine degradation, respectively, while the combination of quercetin and resveratrol reversed some of the differential metabolite changes [68]. In addition, Zhao et al. also reported that a combination of quercetin and resveratrol (2:1) significantly reduced TC, TG, and LDL-C levels in HFD-fed rats [85]. These results suggest that quercetin and resveratrol have significant synergistic hypolipidemic effects.

Park et al. investigated the combined effects of quercetin, resveratrol, and genistein on adipogenesis and apoptosis in human primary adipocytes (HAS) and 3T3-L1 mouse adipocytes (MAS). If these active substances were used to treated HAS alone, lipid accumulation was reduced by 16.8%, 20.3%, and 17.4%, respectively, while combined treatment (2:2:1) reduced lipid accumulation by 80.3%. The combination showed a greater inhibition of lipogenesis compared with the predicted superimposed effect based on individual compounds, indicating a synergistic hypolipidemic effect of a certain proportion of quercetin, resveratrol and genistein combination therapy [69]. Similarly, to assess the synergistic lipid-lowering effects of *Hawthorn* phytochemicals, Huang et al. used a combination of quercetin, hyperoside, rutin, and chlorogenic acid (6:9:2:1). They measured the inhibition of 3-hydroxy-3-methylglutaric acid monoacyl coenzyme A reductase before and after treatment with this combination therapy. The results showed that the inhibition rate of the combination was 58.9% higher than the sum of their individual inhibition rates, indicating that there was indeed a synergistic effect between the four active ingredients [70]. 

In addition, kaempferol, a flavonol in edible plants, has various effects such as antioxidant, anti-inflammatory and hypolipidemic effects, and it can be used as a therapeutic agent for diabetes and cardiovascular diseases [86]. Cinnamaldehyde, a natural flavoring, inhibits glycolysis while enhancing glucose storage [87]. Their combination has been reported to significantly reduce serum TC, TG and LDL-C levels and increase HDL-C levels in mice [71]. The nontargeted metabolomics results also confirmed the simultaneous obstruction of glucose and amino acid metabolism by kaempferol and cinnamaldehyde, showing synergistic hypolipidemic effects.

### 3.2. Synergistic Hypolipidemic Effects of Polysaccharides with Other Categories of Phytochemicals

It has been reported that polysaccharides and polyphenols in green tea can effectively reduce serum leptin levels and inhibit fatty acid absorption in rats, and the combination can reduce lipid accumulation more than their individual effects, which implies that polysaccharides and polyphenols may have synergistic effects in lowering blood lipids [72]. In addition, pumpkin polysaccharides and puerarin both showed lipid-lowering activity by lowering TC, TG and LDL-C levels and improving HDL-C levels [88,89]. Chen et al. investigated the hypoglycemic and hypolipidemic effects of pumpkin polysaccharides and puerarin in the type II diabetes mellitus mice model. After eight weeks of treatment, blood samples were taken from the tail vein of mice that had fasted overnight for the study. The results showed that pumpkin polysaccharide, gerberoside and their combination all improved the blood glucose levels in diabetic mice. Furthermore, compared with pumpkin polysaccharide and puerarin alone, the combination (2:1) treatment more significantly reduced serum TC, TG and LDL-C levels and increased serum HDL-C levels, indicating that they have synergistic hypoglycemic and hypolipidemic potential [73].

In addition, oat β-glucan and phytosterols have been recognized as adjunct or alternative lipid modulating therapies for optimizing dyslipidemia control as they are safe, effective and easily compliable for individuals with dyslipidemia [90,91]. Ferguson et al. reported that oat β-glucan and phytosterols can reduce blood cholesterol levels through different mechanisms and have the potential synergistic hypolipidemic effects. This has also been demonstrated through clinical studies that high molecular weight oat β-glucan and phytosterols have synergistic effects in lowering cholesterol in hypercholesterolemic adults. Specifically, TC and LDL-C decreased significantly by 11.5% and 13.9% (*p* < 0.0001), respectively, after their combined treatment, but they were significantly higher than phytosterols, which were 4.6% and 7.6% (*p* < 0.05), and oat β-glucan, which were 5.7% and 8.6% (*p* < 0.01) [74].

### 3.3. Synergistic Hypolipidemic Effects of Polyphenols with Other Categories of Phytochemicals

Apples are rich in polyphenols and pectin. In order to determine the role of apple components in lipid lowering, mice were fed diets containing 5 g/100 g apple pectin and 10 g/100 g high polyphenol freeze-dried apples or both. The combination was more effective in reducing circulating cholesterol and triglyceride concentrations than feeding alone, suggesting a positive interaction between apple pectin and polyphenols on lipid metabolism [75]. In addition, Ker et al. reported that peeled apples contain a large amount of inositol and uronic acid, which may play a synergistic role in lowering blood lipids, and suggested that phenolics may also have a potential contribution [92].

Epigallocatechin-3-gallate (EGCG) is the main polyphenol in green tea and has high antioxidant, hypolipidemic and anti-inflammatory activities [93,94]. Yang and Zhu et al. found that the combination of EGCG and caffeine was more effective in inhibiting fat accumulation than the same dose alone [95]. In another experiments, they demonstrated that the combination of low-dose EGCG and caffeine (2:1) had synergistic lipid-lowering effects. Specifically, mice were fed with low-dose EGCG (40 mg/kg/d), low-dose caffeine (20 mg/kg/d), high-dose EGCG (160 mg/kg/d), and a combination (40 mg/kg/d EGCG and 20 mg/kg/d caffeine). Compared to single treatment, the combination had more significant effects in reducing hepatic TC and TG levels, preventing weight gain, and inhibiting perirenal and epididymal fat accumulation. In addition, the combination of low-dose EGCG and caffeine resulted in better lipid-lowering effects than high-dose EGCG [76]. Similarly, Sugiura et al. reported that combined treatment with EGCG and caffeine (2:1) had overall stronger inhibitory effects on fat accumulation in mice than either alone [96]. In addition, a recent study reported that the combination of hydrophilic EGCG and lipophilic lycopene synergistically reduced TG and TC levels in serum and liver [77].

Curcuminoid is a natural polyphenol compound, which has good anti-inflammatory and hypolipidemic effects [97,98]. Hasimun et al. evaluated the effects of curcuminoid, S-methyl cysteine and their combination on the regulation of cholesterol levels in serum, liver, and feces. They established an animal model of rats with cholesterol metabolism abnormality induced by propylthiouracil for 7 days. The results showed that curcuminoid, S-methyl cysteine and their combination (1:1) could maintain the normal level of serum cholesterol by inhibiting the absorption of liver cholesterol. Furthermore, the combination resulted in the conversion of cholesterol into feces at a rate three times higher than that of the control group, which was superior to the effect of curcumin and S-methyl cysteine alone. This demonstrated that the combination of curcumin and s-methyl cysteine had synergistic hypolipidemic effects [78].

### 3.4. Synergistic Hypolipidemic Effects of other Different Categories of Phytochemicals

Ursolic acid is a naturally occurring triterpenoid found in many plants which has anti-oxidative, anti-inflammatory and hypolipidemic properties [99]. Artesunate is one of many derivatives of artemisinin extracted from *Artemisia annua*. Researchers investigated the hypolipidemic effects of ursolic acid and artesunate in rabbits with Western-diet induced hyperlipidemia. Rabbits received ursolic acid (25 mg/kg) or artesunate (25 mg/kg) alone or in combination (12.5 + 12.5 mg/kg). The results showed that ursolic acid or artesunate alone significantly reduced plasma triglyceride levels but had no effect on cholesterol levels. The combination reduced triglyceride and cholesterol levels with stronger synergistic effects than their individual effects [79]. This synergistic effect may be attributed to the different hypolipidemic mechanisms of artesunate and ursolic acid [100]. 

In addition, both policosanol and 10-dehydrogingerdione are natural phytochemicals and have shown the ability to lower the level of blood lipids [101,102]. It has been reported that the combination (1:1) significantly decreased serum levels of TC, LDL-C and TG and increased HDL-C levels in mice compared to single treatment, indicating synergistic hypolipidemic effects [80].

## 4. The Synergistic Hypolipidemic Mechanisms

The hypolipidemic effects of phytochemicals are closely related to lipid metabolism disorder, making understanding the formation process of lipid metabolism necessary for the phytochemicals synergistic study. The regulation of lipid metabolism is a complex process involving multiple pathways and targets. Currently, researchers mainly investigate their hypolipidemic mechanism by inhibiting the absorption of exogenous lipids, synthesis of endogenous lipids, and regulating lipid transport and metabolism [103]. As for the regulation of lipid metabolism, it focuses on the regulation of total cholesterol metabolism, but there are few studies on triglyceride. The cholesterol metabolism mainly includes some important molecular mechanisms, such as the increase in reverse cholesterol transport, the inhibition of intestinal cholesterol absorption, the acceleration of liver cholesterol excretion, and the decrease in cholesterol synthesis [14]. According to previous studies, the absorption, synthesis, transport and metabolism of lipids mainly involve the following pathways or targets: niemann-pick protein C1 (NPC1) [104], ATP-binding cassette protein A1 (ABCA1) [105], LDL receptor protein (LDLR) [106], 3-hydroxy-3-methylglutaryl coenzyme A reductase (HMGCR) [107], sterol-regulatory element binding protein 2 (SREBP-2) [108], ATP citrate lyase (ACLY) [109], peroxisome proliferator activated receptor (PPAR) [110], cholesteryl ester transfer protein (CETP) [111], cholesterol-7α-hydroxylase (CYP7A1) [112], fatty acid synthetase (FAS) [113], acetyl-CoA carboxylase (ACC) [114], 5′-monophosphate-activated protein kinase (AMPK) [115], carnitine palmitoyltransferase1A (CPT1A) [116], carbohydrate responsive element-binding protein (ChREBP) [117], hormone-sensitive lipase (HSL), adipose triglyceride lipase (ATGL) [118], and liver x receptor alpha (LXRα) [119]. Figure 4 shows some important targets or pathways for the phytochemicals regulation of lipid metabolism. Figure 5 shows the fatty acid metabolism mechanism and some important targets.

### 4.1. The Synergistic Hypolipidemic Mechanisms Based on the Pathway Analysis

#### 4.1.1. Synergistic Hypolipidemic Mechanisms Based on the Same Pathway

5′-monophosphate-activated protein kinase (AMPK) is regarded as the main energy sensor to maintain the energy homeostasis of cells [90]. The pathways modulated by AMPK are grouped into four general categories, including protein metabolism, lipid metabolism, glucose metabolism, autophagy and mitochondrial homeostasis. The activated AMPK can reduce lipid synthesis by inhibiting the expression of downstream targets (Figure 6) [120,121,122]. Many studies have shown that natural phytochemicals such as resveratrol, epigallocatechin gallate, berberine and quercetin can inhibit lipid metabolism disorders by regulating AMPK activity and its related pathways [123,124]. For example, to investigate the hypolipidemic effects of the combination of kaempferol and cinnamaldehyde, Gao et al. used untargeted metabolomics to reveal the cross-links of metabolic pathways affected by them. Then, the energy state was reflected by indexes such as the contents of adenosine triphosphate (ATP) and adenosine monophosphate (AMP) and the phosphorylation of AMPK. When using the combination of kaempferol and cinnamaldehyde, the synergistic effects will lead to a shift toward catabolic lipid metabolism as the main source of energy supply and an increase in the AMP/ATP ratio, which is due to the hindrance of kaempferol on glycolysis and the effect of cinnamaldehyde on amino acid metabolism. AMPK also acts as an energy receptor of the body, which is activated by an increase in the AMP/ATP ratio. Then, activated AMPK inhibits the synthesis of fatty acids by inhibiting key enzymes such as acetyl-CoA carboxylase, promoting the catabolism of lipids and thus increasing the production of ATP. The results confirmed that the combination of kaempferol and cinnamaldehyde ameliorated glucose and lipid metabolism disorders by enhancing lipid metabolism via the activation of AMPK pathway [71]. 

Furthermore, it has also been reported that the combination of quercetin and resveratrol reversed the HFD-induced inhibition of 5’-adenosine monophosphate activated protein kinase α1 (AMPKα1) phosphorylation and sirtuin 1 (SIRT1) expression in the epididyotic adipose tissue of mice, suggesting that the combination may inhibit obesity and associated inflammation in rats fed with an HFD through the AMPKα1/SIRT1 signaling pathway [125]. Since both resveratrol [126] and quercetin [127] can individually regulate this signaling pathway, the synergistic effect of resveratrol and quercetin at least results from the common AMPKα1/SIRT1 signaling pathway [23]. It has also been reported that the combination can regulate lipid metabolism by inhibiting the activity of ACC [83].

In conclusion, the AMPK pathway is the signal pathway most closely related to the synergistic hypolipidemic effect of phytochemicals.

#### 4.1.2. Synergistic Hypolipidemic Mechanisms Based on Different Pathways

The synergistic hypolipidemic mechanism of phytochemicals in different pathways has been well studied. For example, some studies have revealed the synergistic hypoglycemia mechanism with pumpkin polysaccharides and puerarin through upregulating the expression of critical proteins in the nuclear factor E2 related factor 2 (Nrf2) and phosphoinositide-3-kinase (PI3K) signaling pathways [73].

Oat β-glucan as the main soluble fiber found in oat [128] can increase the excretion of cholesterol by inhibiting bile acid reabsorption [129], while phytosterols reduce the absorption of intestinal cholesterol, which leads to an increased excretion of cholesterol in the feces [130]. Thus, oat β-glucan and phytosterols have two different mechanisms in cholesterol metabolism, which complement each other so effectively that they amplify the reduction in plasma LDL-C [74]. Similarly, Hannan et al. suggested that oligomeric proanthocyanidins, pterostilbene, and niacin have hypolipidemic effects by up-regulating CYP7A1 to induce bile acid secretion and fairly low-density lipoprotein (VLDL) secretion from the liver, up-regulating PPAR to promote cholesterol metabolism, and inhibiting cellular TG and free fatty acid synthesis, respectively. The synergistic lipid-lowering effects can be observed with the combination of the three, and low-dose mixed treatment has been proved more effective than a single high dose [35]. 

In addition, combined treatment with crocin, chlorogenic acid, geniposide and quercetin increased ABCA1, CYP7A1 and AMP-activated protein kinase 2α(AMPKα2) mRNA expression while decreasing SREBP2 and LXRα mRNA expression [24]. This result suggests that their synergistic hypolipidemic effects seem to be achieved through different pathways, such as the regulation of AMPK, LXRα and involvement during cholesterol synthesis and metabolism. 

Nowadays, EGCG is the most widely used in the studies on the synergistic lipid-lowering mechanism of phytochemicals. Zhu et al. proposed that the combination of EGCG and caffeine could target different lipid-lowering pathways, which may be the basis for their synergy [76]. They also found significant synergistic effects from many aspects of the combination-treated mice, such as increased fecal acetate, propionic acid and short-chain fatty acids leading to decreased expression of G protein-coupled receptors (GPRs) and increased fecal bile acid loss. Furthermore, combined treatment with EGCG and caffeine could have synergistic effects on increasing hepatic G protein-coupled bile acid receptor 1 (TGR5) expression and decreasing intestinal farnesoid X receptor (FXR) and fibroblast growth factor 15 (FGF15) expression, leading to increased hepatic CYP7A1 expression. It has also been reported that the combination of catechins and caffeine can inhibit fat accumulation by suppressing fatty acid synthesis and up-regulating enzymatic activities involved in β-oxidation of fatty acid in the liver, but the combination of EGCG and caffeine did not show the same effect [96]. In addition, Wang et al. demonstrated that the combination of EGCG and lycopene could synergistically lower lipids through different pathways, including HMGCR, LDLR, PPAR, and AMPK at both mRNA and protein levels, and the synergistic hypolipidemic effects could be achieved mainly through the activation of the AMPK pathway [77].

In addition, sea buckthorn flavonoids have been shown to be a potential nutrient for the prevention of cognitive impairment caused by high-energy-density diets, because they can activate different signaling pathways to inhibit inflammation and regulate lipid metabolism [131]. Network pharmacology analysis indicated that sea buckthorn flavonoids improved hyperlipidemia by regulating multiple pathways, such as cholesterol metabolism, fat digestion and absorption, the PPAR signaling pathway, the AMPK signaling pathway, and insulin resistance [32]. Specifically, quercetin, hesperidin, and catechin induced LXRα, ABCA1, and apolipoproteins A1 expression to increase cholesterol efflux, quercetin, hesperidin, catechin and isorhamnetin inhibited SREBP-2 and its target gene LDLR to decrease cholesterol de novo synthesis, and hesperidin and catechin up-regulated CPT1A to accelerate fatty acids oxidation (Figure 7). These results suggest that the effects of sea buckthorn flavonoids on improving hyperlipidemia are actually the synergistic effects of different phytochemicals.

### 4.2. Target-Based Analysis of Synergistic Mechanisms

#### 4.2.1. Synergistic Hypolipidemic Effects Based on the Same Target

Different phytochemicals with similar structures can be combined with the same target to produce superimposed or enhanced synergistic hypolipidemic effects. For example, 3-hydroxy-3-methylglutaryl coenzyme A reductase (HMGCR) is a regulatory enzyme involved in liver cholesterol biosynthesis, which is the target enzyme of anti-hyperlipidemia drugs [132,133]. The commonly used statins achieve its lipid-lowering effect through inhibiting HMGCR [134]. Quercetin, hyperoside, rutin and chlorogenic acid from hawthorn have been reported to inhibit HMGCR activity, and there are synergistic effects between the monomers after the combination [70]. Furthermore, Susilowati et al. demonstrated that quercetin, chlorogenic acid, epicatechin and catechin from apple peel have inhibitory effects on HMGCR targets by molecular docking tests and in vitro experiments, leading to a combined inhibition of cholesterol synthesis [135].

Cholesterol ester transfer protein (CETP) plays a significant role in high-density lipoprotein metabolism and reverse cholesterol transport [111,136]. The results showed that both policosanol and 10-dehydrogingerdione could prevent hyperlipidemia by inhibiting the activity of the CETP target, reducing serum TG level and increasing HDL-C content [137,138], and the combination of these two phytochemicals could lead to a further enhancement of HDL-C elevation [80]. In our previous study, we found that both biochanin A and chickpea peptide Cpe-III could exert hypolipidemic effects based on the target CETP [139]. Further investigation has demonstrated that the synergistic mechanism between these two may be related to the regulation of gene expression related to lipid synthesis, metabolism, and oxidation [140].

#### 4.2.2. Synergistic Hypolipidemic Effects Based on Multi-Target

Synergistic multi-target effects refer to the synergistic effect of multiple phytochemicals on different targets, which is considered as a mechanism of synergistic effect [141]. Combined phytochemicals can target different biomarkers, resulting in synergistic hypolipidemic effects, such as direct interactions with different target proteins in the same metabolic pathway [22]. The active components in the combination may affect several targets in the hypolipidemic pathway, such as transporter proteins, receptors, and enzymes. Fatty acid synthase (FAS) is considered as the key enzyme to catalyze fatty acid synthesis [142]. Sterol regulatory element-binding protein-1c (SREBP-1c) mainly regulates the biosynthesis of fatty acids, triglycerides and cholesterol [143,144]. Li et al. found that the expression of SREBP-1c and FAS reduced by 1.67-fold and 0.62-fold, respectively, in mice fed with the combination of soluble dietary fiber and insoluble dietary fiber compared with the control group, suggesting that the combination could regulate fatty acid synthesis by down-regulating the expression of FAS and SREBP-1c [34]. 

Peptide transporter 1 (PepT1) is one of the key targets for the absorption of hypolipidemic drugs into the bloodstream, and it is mainly found in the small intestine [145]. Peroxisome proliferator-activated receptor alpha (PPARα) is a classical lipid-lowering target, and PPARα agonists can exert lipid-lowering effects by regulating the metabolic pathways of blood lipids [146]. PPARα has been found to regulate the expression of various intestinal transporters, especially PepT1, and some PPARα agonists have shown the ability to promote the uptake of PepT1 substrates [147]. Qiao et al. suggested that the combination of Panax notoginseng and Salvia miltiorrhiza containing active components could activate both PepT1 and PPARα targets, which may be the basis for their synergistic hypolipidemic effects [148].

In addition, Shimizu et al. also found that the combination of flavonoids contained in Scutellaria root inhibited inflammation better than using flavonoids alone, and the synergistic effect of the flavonoid combination was actually the sum of the effects of the flavonoids inhibiting different targets [149]. Furthermore, Ma et al. selected the combination of acacetin and apigenin (1:1) from 12 flavonoid compounds of Artemisia sacrorum. The combination showed significant synergistic inhibition of various genes or proteins related to lipid synthesis (SREBP1c, FAS, PPAR), and this combination synergistically promoted the phosphorylation of AMPK and ACC1 [30].

### 4.3. Other Mechanisms

The hypolipidemic interactions between different phytochemicals may increase their bioavailability in vivo [26]. Therefore, synergistic effects based on improved solubility and absorptivity to enhance the bioavailability of phytochemicals have drawn the interest of researchers [141]. The bioavailability of phytochemicals is a critical factor for functional foods and health claims related to food ingredients, leading to an understanding of the mechanisms of action associated with benefits [21]. For example, studies have shown a 154% increase in the bioavailability of curcumin in rats given both piperine and curcumin (10:1) and a staggering 2000% increase in humans treated with both [150]. This is most likely the result of piperine inhibiting the glucuronidation of curcumin, as curcumin is fully metabolized in the form of glucuronic acid before reaching the plasma [151]. Zhao et al. found that the bioavailability of quercetin increased in the presence of proanthocyanidins, which was possibly because it effectively improved the chemical stability of quercetin by preventing quercetin oxidation and increasing solubility [152]. In addition, a study reported that the substantial interaction between cocoa flavanols and methylxanthines existed at the level of absorption, in which the methylxanthines mediated an increased plasma concentration of epicatechin metabolites and synergistically enhanced the anti-hypertensive effect of epicatechin [153].

In addition, a recent study reported that EGCG and caffeine exhibited synergistic hypolipidemic effects in altering gut microbiota, including decreased Firmicutes level and increased Bifidobacterium level in mice [76]. This synergistic effect may be due to the regulation of the gut microbiota and bile acid metabolism by the combination treatment.

## 5. Conclusions and Future Directions

The phytochemicals in food are diverse in variety, numerous in structure, and complexly interact with each other. As an effective initiative to protect and improve the activity of phytochemicals, the concept of “synergy” and its application are highly sought after in the development of hypolipidemic functional foods. Based on the complexity of the regulatory network of lipid metabolism, synergistic therapy has shown to be superior to monotherapy [154]. There is a lot of evidence that plant extracts show better results than isolated individual phytochemicals in lipid-lowering. For example, Hannan et al. reported that low-dose combination therapy was more effective in lipid-lowering than individual bulk dose in rabbits fed with a high-cholesterol diet [35]. Single lipid-lowering substances, whether drugs or natural active substances, are increasingly showing their own limitations in practical application. On the one hand, a single substance given in high doses may become toxic rather than beneficial. On the other hand, the effect of lipid-lowering levels was significant at lower doses, but this effect was not as significant as predicted at higher doses, indicating a ceiling effect. 

Phytochemicals are an important source of therapeutic agents for hyperlipidemia. In particular, synergistic interactions between phytochemicals provide an interesting approach to achieve lipid-lowering effect in human. The regulation process of lipid metabolism and multi-target intervention by phytochemicals occurs during the absorption, synthesis, transport and metabolism of lipids, and synergistic regulatory effects may be essential in regulating these pathways. HMGCR, LDLR, AMPK, ABCA1, CYP7A1, CETP, SREBP and PPAR remain the key molecules involved in these synergistic processes. In addition, there are a variety of phytochemicals that have been shown hypolipidemic effects. However, studies of synergistic effects have been focused on flavonoids, polysaccharides, and phenols in the phytochemicals. In general, phytochemicals may work together to prevent hyperlipidemia by regulating different metabolic pathways or targets.

There are still some urgent issues to be solved. For example, although some plant extracts or foods have been shown to have synergistic hypolipidemic effects, the structure of most phytochemicals and their structure–activity relationships have not been determined due to limitations in research methods and techniques, which makes it difficult to study the mechanism of synergy. In fact, the study on hytochemicals synergy is closely related to its own molecular structure. For example, quercetin is a flavonoid compound with a variety of active structures [155]. These special active structures make it possible to create synergies with other phytochemicals, and phytochemicals with similar structure may have similar, overlapping or complementary roles in their lipid-lowering activities. However, suitable methods to understand these interactions are needed. Molecular simulation and molecular docking are considered to be important methods for the discovery and design of synergistic interactions between phytochemicals due to their advantages of rapidity and high accuracy [156,157]. Molecular simulation and molecular docking reveal binding information between candidate molecules and enzymes and obtain interaction trajectories to better understand biological processes, which is considered as a general approach to study the interactions and conformational stability of biomolecules [158]. In addition, the application of network pharmacology and metabolomics has also made a positive contribution to the discovery of synergistic active components in plants. Network pharmacology can transform the study of disease from a “one-target, one-drug” model to a “network-target, multi-component-treatment” model [159]. Metabolomics can be used to elucidate changes in metabolites following endogenous or exogenous perturbations in vivo and to search for biomarkers and pathways in plasma and liver samples [160,161], which are important for screening active compounds in plants, discovering targets and exploring mechanisms of action. For example, based on the results of a network pharmacology and metabolomics studies, Wang et al. found quercetin and aloin as potential active ingredients that may exert synergistic efficacy in the treatment of hyperlipidemia [162]. In short, in order to study the synergistic hypolipidemic effects among phytochemicals better, we should establish a comprehensive model, which should have the roles of screening of synergistic substances, quantification of synergistic effects, and evaluation of synergistic mechanisms.

In conclusion, increasing evidence suggests that phytochemical combinations are a practical approach to preserve and improve the health-promoting effects of functional ingredients. The concept of synergistic hypolipidemic provides a theoretical basis for the development of phytochemicals with synergistic effects for the treatment of hyperlipidemia and its complications. It is expected that improvements in synergistic research models and complex mixture analysis techniques will help us to better explore the synergistic therapeutic potential between phytochemicals in the near future. The concept of synergistic effects of phytochemical combinations is also a guideline for the design of functional foods for hypolipidemic.

## Figures and Tables

**Figure 1 foods-11-02774-f001:**
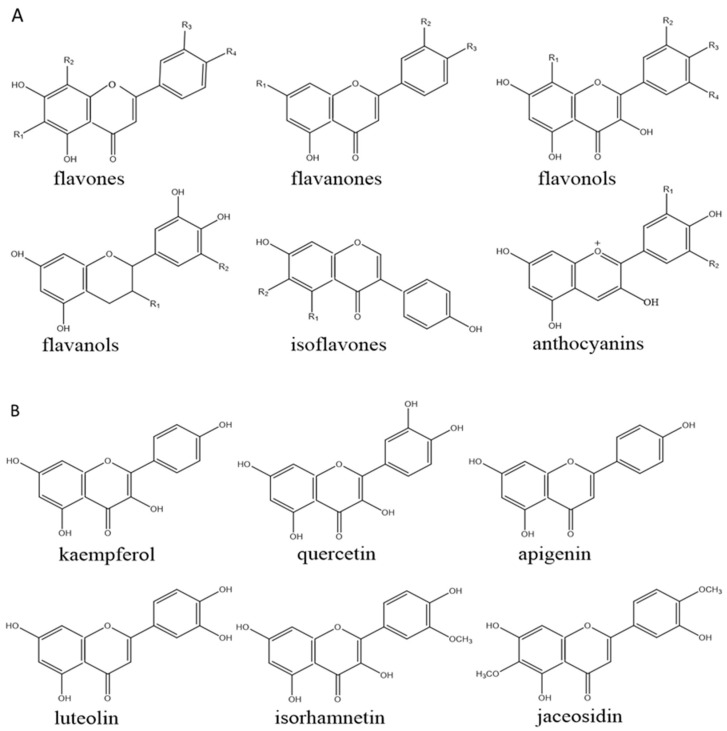
(**A**) Basic structures of major flavonoids. (**B**) Basic structures of flavonoids that have been found to have possible synergistic hypolipidemic effects.

**Figure 2 foods-11-02774-f002:**
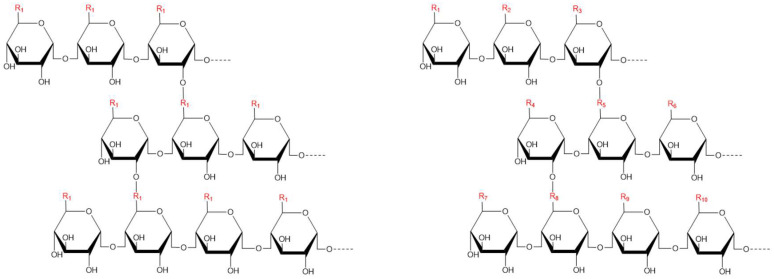
Two types of polysaccharides: homo-polysaccharides and hetero-polysaccharides.

**Figure 3 foods-11-02774-f003:**
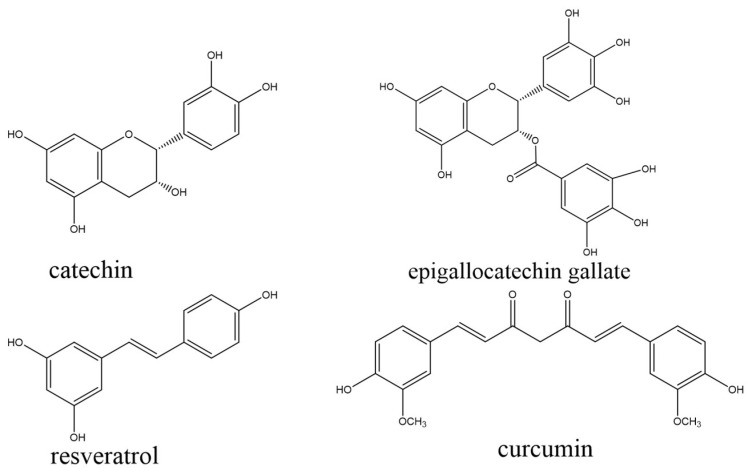
Basic structures of polyphenols that have been found to have possible synergistic hypolipidemic effects.

**Figure 4 foods-11-02774-f004:**
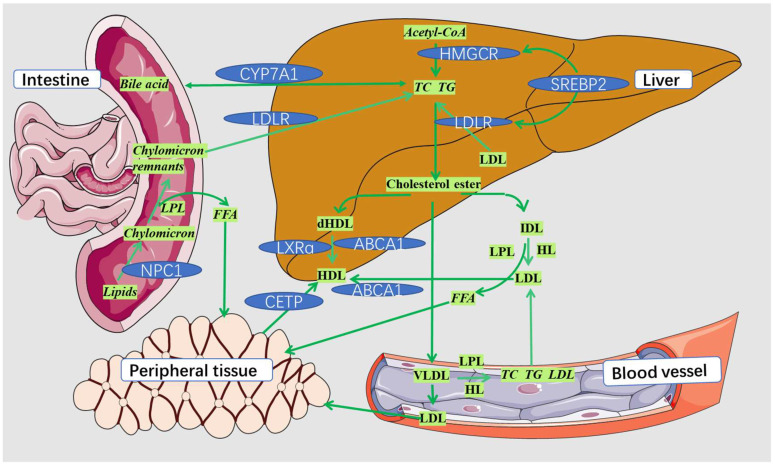
Targets or pathways for phytochemicals regulation of lipid metabolism. Lipoprotein lipase (LPL), hepatic lipase (HL), low-density lipoprotein (LDL), intermediate-density lipoprotein (IDL), high-density lipoprotein (HDL), free fatty acids (FFA).

**Figure 5 foods-11-02774-f005:**
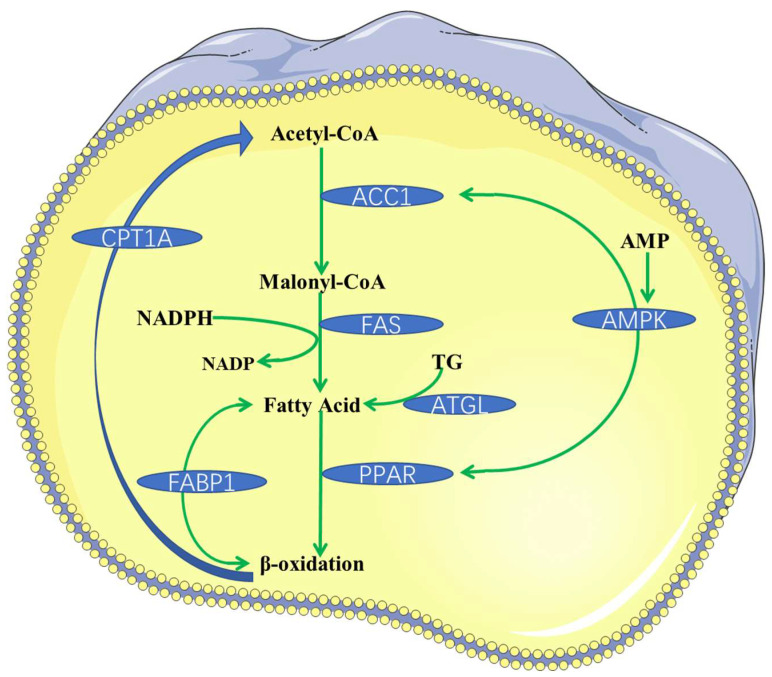
Fatty acid metabolism mechanism and important targets.

**Figure 6 foods-11-02774-f006:**
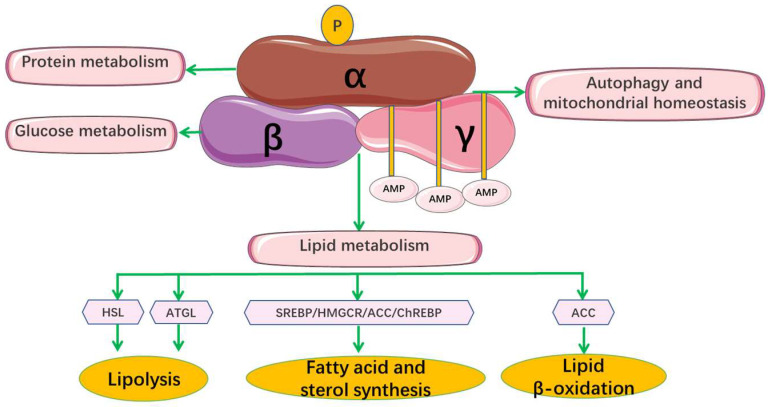
AMPK’s structural diagram and regulation of various metabolic processes.

**Figure 7 foods-11-02774-f007:**
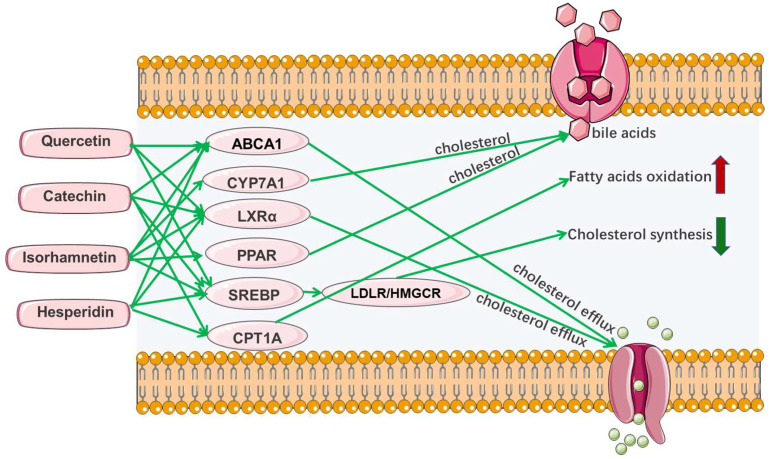
The hypolipidemic mechanism of sea buckthorn flavonoids speculated by network pharmacology.

**Table 1 foods-11-02774-t001:** Experimental assessment and mechanism of synergistic hypolipidemic effects between phytochemicals of the same category found in plants.

Category	Phytochemical Combinations	Ratios and Concentration	Experimental Model	Effect or Mechanisms	Reference
Flavonoids	quercetin and kaempferol	1:1, 15 μM each	HepG2 cells	Improved LDL-C uptake more effectively	[29]
Flavonoids	acacetin and apigenin	1:1, 10 μM each	3T3-L1 cells	Promoted the phosphorylation of AMPK and ACC	[30]
Flavonoids	rutin and epicatechin	1:3, 10 mg/kg, 30 mg/kg	Alloxan-induced diabetic mice	Showed a very significant improvement in body weight and a potent antihyperglycemic activity	[31]
Flavonoids	quercetin, catechin, hesperidin and isorhamnetin	2:2:2:1, 40 μM, 20 μM	HL7702 cells	Down-regulated the mRNA expression of SREBP-2 and LDLR.	[32]
Polysaccharides	high molecular weight dextran and low molecular weight heteropolysaccharide	1:1, 50 μg/mL each	RAW264.7 macrophage cell	Demonstrated stronger inhibitory effect on NO, TNF- α, and IL- 6 production	[33]
Polysaccharides	soluble dietary fiber and insoluble dietary fiber	1:1, 0.15 g/kg each	Sprague–Dawley rats	The mRNA expression levels of lipid synthesis genes SREBP-1c and FAS were significantly down-regulated	[34]
Polyphenols	oligomeric proanthocyanidins and pterostilbene	5:3, 50 mg/kg, 30 mg/kg	Male albino rabbits	The LDL/HDL ratio and atherogenic index were suppressed by 59.3% and 25%	[35]
Polyphenols	catechin, hesperidin, ferulic acid and quercetin	20:9.3:4.3:2 20, 9.3, 4.3, 2 μMol/L	Human plasma	Effects of polyphenols protecting LDL from oxidation were observed	[36]
Amides	zanthoxylum and capsaicin	3:6 3 mg/kg, 6 mg/kg	Sprague–Dawley rats	Reduced the serum levels of TC, TG, and LDL-C	[37]

**Table 2 foods-11-02774-t002:** Experimental assessment and mechanism of synergistic hypolipidemic effects between different categories of phytochemicals found in plants.

Category	Phytochemical Combinations	Ratios and Concentration	Experimental Model	Effect or Mechanisms	Reference
Flavonoids with Polyphenols	quercetin and resveratrol	2:1, 30 mg/kg/d, 15 mg/kg/d	Rats fed an HFD	May suppress obesity and associated inflammation via the AMPKα1/SIRT1 signaling pathway	[68]
Flavonoids with polyphenols	genistein, quercetin and resveratrol	1:2:2, 6.25 Μ_M_, 12.5 Μ_M_, 12.5 μM	Human primary adipocytes and 3T3-L1 mouse adipocytes	Lipid accumulation was reduced by 80.3%; resulted in a significant decrease in lipid accumulation	[69]
Flavonoids with polyphenols and terpenoids	quercetin, crocin, chlorogenic acid and geniposide	10:1:30:10 10 μmol/L,1 μmol/L, 30 μmol/L, 10 μmol/L	HepG2 cells	Increased ABCA1, CYP7A1, and AMPKα2 mRNA expression, decreased SREBP2, and LXRα mRNA expression	[24]
Flavonoids with polyphenols	quercetin, hyperoside, rutin and chlorogenic acid	6:9:2:1	Inhibitory activity of HMG-CoA reductase	Increased the inhibitory activity of HMG-CoA reductase by 58.9%	[70]
Flavonoids with aldehydes	kaempferol and cinnamaldehyde	39: 58, 39 mg/kg, 58 mg/kg	Eight-week-old male Kunming mice	Ameliorated glucose and lipid metabolism disorders by enhancing lipid metabolism via the activation of AMPK	[71]
Polysaccharides with polyphenols	tea polysaccharide and polyphenols	1:1, 400 mg/ kg each	Sprague–Dawley rats fed with high-fat diet	Reduced rat serum leptin levels, inhibited the absorption of fatty acids, reduced the expression levels of IL-6, TNF-α gene	[72]
Polysaccharides with flavonoids	pumpkin polysaccharides and puerarin	2:1, 400 mg/kg, 200 mg/kg	Male Kunming mice	Up-regulated the expression of the critical proteins in the Nrf2/HO-1 and PI3K/Akt signaling pathways.	[73]
Polysaccharides with phytosterols	oat β-glucan and phytosterols	3:2, 3 g/d, 2 g/d	Healthy adults aged 18–70 years old	TC: HDL-C ratio was significantly reduced	[74]
Polyphenols with polysaccharides	pectin and polyphenols	1:2 5 g/100 g, 10 g/100 g	Male Wistar rats	Significantly lowered plasma cholesterol and triglycerides	[75]
Polyphenols with alkaloids	epigallocatechin-3-gallate and caffeine	2:1, 40 mg/kg/d,20 mg/kg/d	Four-week-old Sprague–Dawley male rats	Improved gut microbiota, inhibited fat accumulation, increased expression of hepatic TGR5	[76]
Polyphenols with carotenoids	epigallocatechin-3-gallate and lipophilic lycopene	3:1, 30 mg/kg, 10 mg/kg	Healthy 4-week-old male Sprague–Dawley rats	Triggered the pathways of HMGCR, LDLR, PPAR and AMPK	[77]
Polyphenols with amino acids	curcuminoid, S-methyl cysteine	1:1, 50 mg/kg each	Rats with cholesterol metabolism abnormality	Increased the conversion of cholesterol into the feces as much as 3 times	[78]
Terpenoids with acetyl compounds	ursolic acid and artesunate	1:1, 12.5 mg/kg each	Rabbit fed with Western-type diet	Significantly decreased the plasma cholesterol and triglyceride	[79]
Others	policosanol and 10-dehydrocongerdione	1:1, 10 mg/kg each	Adult male albino rabbits	Resulted in a CETP inhibitory activity, increasing HDL-C level	[80]

## Data Availability

Not applicable.

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
