# Peer review of "Synergistic Hypolipidemic Effects and Mechanisms of Phytochemicals: A Review"

_foods, 2022, doi:10.3390/foods11182774_

Round 1
Reviewer 1 Report
The manuscript topic is interesting and will contribute to the field. However, some corrections are needed.
Authors are referring to the structural complexity of phytochemicals in foods as “Phytochemicals in food have complicated chemical structures and complex interactions.” or “The complex chemical components of phytochemicals facilitate the combined action on multiple cellular and molecular targets, leading to measurable biological effects.”
In general, phytochemicals that are mentioned in this paper are dominantly small molecules, that we cannot define as typically ‘complex’. On the other hand, the composition of phytochemicals in foods are rather ‘complex’ – regarding the number and variety of compound present.
Division to ‘flavonoids’ and ‘polyphenols’ and consequent discussion for synergic effects in between the molecules from the ‘same’ and ‘different’ category of molecules should be revisited and clarified. As all polyphenols (comprising various subgroups) are observed as one category, while flavonoids are a separate category. The approach is clear but the explanation and naming should be more precise and accurate.
“For example, quercetin is both a flavonoid and a phenolic compound with a variety of active structures” is rather an unusuall statement, taking into account that all flavonoids are phenolic compounds. Furthermore,
English language should be extensively checked and edited, by a native/proficiency speaker, with considerable knowledge of the topic. Due to language errors, some phrases are senseless, eg.: “Food intake is also generally a number of substances into the body at the same time.” “It is still a challenge to clarify the mechanism of phytochemicals synergy, especially the biological factors such as bioavailability are often neglected.” Etc…
Author Response
Reviewer #1:
General comments:
The manuscript topic is interesting and will contribute to the field. However, some corrections are needed.
Response: Thank you for your comment. Thank you for your approval of our manuscript. We have revised the article according to your suggestions, and we will answer all your questions below.
Specific comments:
- Authors are referring to the structural complexity of phytochemicals in foods as “Phytochemicals in food have complicated chemical structures and complex interactions.” or “The complex chemical components of phytochemicals facilitate the combined action on multiple cellular and molecular targets, leading to measurable biological effects.” In general, phytochemicals that are mentioned in this paper are dominantly small molecules, that we cannot define as typically ‘complex’. On the other hand, the composition of phytochemicals in foods are rather ‘complex’ – regarding the number and variety of compound present.
Response: Thank you for your question. We have made corresponding modifications to the above-mentioned issues to make our views more rigorous. Specifically, we have modified “Phytochemicals in food have complicated chemical structures and complex interactions” to “The phytochemicals in food are diverse in variety, numerous in structure, and complexly interact with each other”, “The complex chemical components of phytochemicals facilitate the combined action on multiple cellular and molecular targets, leading to measurable biological effects” to “The complex interactions of phytochemicals facilitate the combined action on multiple cellular and molecular targets, leading to measurable biological effects”.
- Division to ‘flavonoids’ and ‘polyphenols’ and consequent discussion for synergic effects in between the molecules from the ‘same’ and ‘different’ category of molecules should be revisited and clarified. As all polyphenols (comprising various subgroups) are observed as one category, while flavonoids are a separate category. The approach is clear but the explanation and naming should be more precise and accurate. “For example, quercetin is both a flavonoid and a phenolic compound with a variety of active structures” is rather an unusuall statement, taking into account that all flavonoids are phenolic compounds.
Response: Thank you for your question. We have reviewed the literature again and made corrections to the contents of the manuscript. Phenolic compounds have at least one aromatic ring with one or more hydroxyl groups attached [1]. Flavonoids are composed of a common diphenylpropane (C6-C3-C6) skeleton in which two aromatic rings are linked by a three-carbon chain [2]. We have modified “quercetin is both a flavonoid and a phenolic compound with a variety of active structures” to “quercetin is a flavonoid compound with a variety of active structures”.
References:
- Del Rio, D.; Rodriguez-Mateos, A.; Spencer, J.P.E.; Tognolini, M.; Borges, G.; Crozier, A. Dietary (Poly)phenolics in Human Health: Structures, Bioavailability, and Evidence of Protective Effects Against Chronic Diseases. Antioxidants & Redox Signaling, 2013. 18(14): p. 1818-1892.
- Saito, K.; Yonekura-Sakakibara, K.; Nakabayashi, R.; Higashi, Y.; Yamazaki, M.; Tohge, T.; Fernie, A.R. The flavonoid biosynthetic pathway in Arabidopsis: Structural and genetic diversity. Plant Physiol Bioch, 2013. 72: p. 21-34.
- Furthermore, English language should be extensively checked and edited, by a native/proficiency speaker, with considerable knowledge of the topic. Due to language errors, some phrases are senseless, eg.: “Food intake is also generally a number of substances into the body at the same time.”, “It is still a challenge to clarify the mechanism of phytochemicals synergy, especially the biological factors such as bioavailability are often neglected.” Etc…
Response: Thank you for your question. Based on your comments, we have asked native English experts to help us improve the language of the article, and we have made a number of changes to the article.
â‘ “Food intake is also generally a number of substances into the body at the same time” to “Food intake is also generally several substances in the body at the same time”.
â‘¡“It is still a challenge to clarify the mechanism of phytochemicals synergy, especially the biological factors such as bioavailability are often neglected” to “Although research relevant to interactive effects among the phytochemicals mounted up, the mechanism of phytochemicals synergy is still not clear. Especially, biological influence factors are often neglected”.
â‘¢“Similarly, in order to evaluate the synergistic hypolipidemic effects of phytochemicals in hawthorn, a combination of quercetin, hyperoside, rutin and chlorogenic acid (6:9:2:1) was used, and then the inhibition of 3-hydroxy-3-methylglutaric acid monoacyl coenzyme A reductase was measured before and after the combination” to “Similarly, to assess the synergistic lipid-lowering effects of Hawthorn phytochemicals, Huang et al. used a combination of quercetin, hyperoside, rutin, and chlorogenic acid (6:9:2:1). They measured the inhibition of 3-hydroxy-3-methylglutaric acid monoacyl coenzyme A reductase before and after treatment with this combination therapy”.
â‘£“In addition, sea buckthorn flavonoids have been shown to be a potential nutrient for the prevention of cognitive impairment induced by high energy density diet because they could activate different signaling pathways to improve inflammation and regulate lipid metabolism” to “In addition, seabuckthorn flavonoids have been shown to be a potential nutrient for the prevention of cognitive impairment caused by high-energy-density diets. Because they can activate different signaling pathways to inhibit inflammation and regulate lipid metabolism”.
⑤“These specific active structures make it possible to create synergistic effects with other phytochemicals, and structurally similar phytochemicals may have similar, overlapping or complementary effects in their hypolipidemic activity” to “These special active structures make it possible to create synergies with other phytochemicals. And phytochemicals with similar structure may have similar, overlapping or complementary roles in their lipid-lowering activities”, Etc…(see the revised version, where the changes have been marked with a revision pattern). Thank you again for your correction.
Please see the attachment.

Reviewer 2 Report
This article is informative and well written.
There are a few points that should be corrected, as follows:
1. P2 L11: phytochemicals
2. P2 L40: phytochemicals
3. P2 L49: showed
4. P2 L49-50 and L53: Hippophae rhamnoides and Zingiber mioga are the scientific names of the plants. They should be italic.
5. P3 L1: Yari et al.
6. P3 L7,8,13: phytochemicals
7. P4 L23,28: kaempferol
8. P4 L33: Artemesia sacrorum should be italic
9. P5 L5: we
10. P6 L33: The studies
11. P7 L22: et al.
12. P7 L27: polyphenols
13. P8 Under 2.4 Synergistic hypolipidemic effects of other phytochemicals (L4 of this paragraph): according to Chen et al. (37), please ensure that this article used 'zanthoxylum and capsaicin' or numb-testing components of Zanthoxylum bungeanum.
14. P12 L40: Artemisia annua should be italic.
15. P16 L5, 23/ P17 L3/ P19 L25,27: phytochemicals
16. References 1, 2, and 12: please recheck the references correctly.
17. References 4, 5, and 111: please shorten the reference and cite the first ten authors, then add a semicolon and add 'et al.' at the end.
Author Response
Reviewer #2:
General comments:
This article is informative and well written.
Response: Thank you for your comment. Thank you for your approval of our manuscript. We have revised the article according to your suggestions.
Specific comments:
There are a few points that should be corrected, as follows: 1-17.
Response: Thank you for pointing out the above issues that appear in the manuscript. We have revised each of them according to your suggestions.
- P2 L11: Phytochemicals to phytochemicals.
- P2 L40: Phytochemicals to phytochemicals.
- P2 L49: Showed to showed.
- P2 L49-50 and L53: Hippophae rhamnoides and Zingiber mioga to Hippophae rhamnoides and Zingiber mioga.
- P3 L1: Yari, et al. to Yari et al.
- P3 L7,8,13: Phytochemicals to phytochemicals.
- P4 L23,28: Kaempferol to kaempferol.
- P4 L33: Artemesia sacrorum to Artemesia sacrorum.
- P5 L5: We to we.
- P6 L33: The study to The studies.
- P7 L22: et al to et al.
- P7 L27: Polyphenols to polyphenols
- P8 Under 2.4 Synergistic hypolipidemic effects of other phytochemicals (L4 of this paragraph): according to Chen et al. (37), please ensure that this article used 'zanthoxylum and capsaicin' or numb-testing components of Zanthoxylum bungeanum.
Response: Thank you for your question. After reviewing the data, we confirmed that the article used “numb-tasting components of Zanthoxylum bungeanum and capsaicin” [1]. Thank you again for your correction.
- P12 L40: Artemisia annua to Artemisia annua.
- P16 L5, 23/ P17 L3/ P19 L25,27: Phytochemicals to phytochemicals.
- References 1, 2, and 12: please recheck the references correctly.
Response: Thank you for your question. According to your comments, we have revised reference 1 to “Boden, W.E.; Bhatt, D.L.; Toth, P.P.; Ray, K.K.; Chapman, M.J.; Luscher, T.F. Profound reductions in first and total cardiovascular events with icosapent ethyl in the REDUCE-IT trial: why these results usher in a new era in dyslipidaemia therapeutics. Eur Heart J, 2020. 41(24): p. 2304”, reference 2 to “Nelson, R.H. Hyperlipidemia as a Risk Factor for Cardiovascular Disease. Primary Care, 2013. 40(1): p. 195”, reference 4 to “Mach, F.; Baigent, C.; Catapano, A.L.; Koskina, K.C.; Casula, M.; Badimon, L.; Chapman, M.J.; De Backer, G.G.; Delgado, V.; Ference, B.A.; et al. 2019 ESC/EAS guidelines for the management of dyslipidaemias: Lipid modification to reduce cardiovascular risk. Atherosclerosis, 2019. 290: p. 140-205”.
- References 4, 5, and 111: please shorten the reference and cite the first ten authors, then add a semicolon and add 'et al.' at the end.
Response: According to your comments, we have revised reference 4 to “Mach, F.; Baigent, C.; Catapano, A.L.; Koskina, K.C.; Casula, M.; Badimon, L.; Chapman, M.J.; De Backer, G.G.; Delgado, V.; Ference, B.A.; et al. 2019 ESC/EAS guidelines for the management of dyslipidaemias: Lipid modification to reduce cardiovascular risk. Atherosclerosis, 2019. 290: p. 140-205”, reference 5 to “Fernandez-Friera, L.; Penalvo, J.L.; Fernandez-Ortiz, A.; Ibanez, B.; Lopez-Melgar, B.; Laclaustra, M.; Oliva, B.; Mocoroa, A.; Mendiguren, J.; de Vega, V.M.; et al. Prevalence, Vascular Distribution, and Multiterritorial Extent of Subclinical Atherosclerosis in a Middle-Aged Cohort The PESA (Progression of Early Subclinical Atherosclerosis) Study. Circulation, 2015. 131(24): p. 2104-2113”, and reference 111 to “Ference, B.A.; Kastelein, J.J.P.; Ginsberg, H.N.; Chapman, J.; Nicholls, S.J.; Ray, K.K.; Packard, C.J.; Laufs, U.; Brook, R.D.; Oliver-Williams, C.; et al. Association of Genetic Variants Related to CETP Inhibitors and Statins With Lipoprotein Levels and Cardiovascular Risk. JAMA-J Am Med Assoc, 2017. 318(10): p. 947-956”.
References:
1.Chen, Z.; Liu, Y.; Lu, H.; Guo, T.; Liu, X. Hypolipidemic Effects of Numb-Tasting Components of Zanthoxylum bungeanum Combined with Capsaicin at Various Ratios on Rats. Food Science, 2014. 35(19): p. 231-235.

Reviewer 3 Report
In the manuscript there are some points that must be clarified :
1. I strongly suggest authors to introduce more keywords. The usefulness of keywords is to make the article both more and more easily searchable visible after its publication through commonly used search engines.
2. The introduction is interesting, but in my opinion, it does not fully cover the topic.
3.In my opinion, the basic structures of reported other phytochemicals can be added in supplementary material, not in the manuscript.
I believe that the work presented for review is of a high technical level, but it requires substantive amendments (please post new items).
The research and the prepared article are of a very high substantive level, based only on the latest scientific reports from the last few years.
I am asking for a deeper description, taking into account my suggestions above, with post new items.
Author Response
Reviewer #3:
General comments:
I believe that the work presented for review is of a high technical level, but it requires substantive amendments (please post new items). The research and the prepared article are of a very high substantive level, based only on the latest scientific reports from the last few years. I am asking for a deeper description, taking into account my suggestions above, with post new items.
Response: Thank you for your comment. Thank you for your approval of our manuscript. We have revised the article according to your suggestions, and we will answer all your questions below.
Specific comments:
1.I strongly suggest authors to introduce more keywords. The usefulness of keywords is to make the article both more and more easily searchable visible after its publication through commonly used search engines.
Response: Thank you for your comments. Based on your suggestion, we have added “combination” and “lipid metabolic pathways” keywords. Thank you again for your correction.
- The introduction is interesting, but in my opinion, it does not fully cover the topic.
Response: Thank you for your comments. Based on your suggestion, to make the introduction fully cover the topic, we have added the paragraph of lipid metabolic pathways in the introduction. “Lipid metabolism is a complex process that involves lipid biosynthesis, absorption, transport, and elimination [1]. Due to the complexity of lipid metabolism regulatory network, synergistic regulation of different metabolic pathways or targets may be more effective than single pathway or target in the treatment of hyperlipidemia. In recent years, the process of lipid metabolism has become more and more clear, and some important pathways and targets have been discovered by researchers. For example, the 5 '-Adenosine monophosphate (AMP)-activated protein kinase (AMPK) is a natural energy sensor in mammalian cells that plays a key role in lipid metabolism [2]”. This lays the foundation for us to study the synergistic lipid-lowering effects of phytochemicals.
In addition, to make the introduction not too lengthy, we have also removed some contents, for example, “Recent research has shown that biologically active phytochemicals in plants can produce a wide range of synergistic hypolipidemic effects”. Thank you again for your correction.
References:
1. Ul Islam, S.; Ahmed, M.B.; Ahsan, H.; Lee, Y.S. Recent Molecular Mechanisms and Beneficial Effects of Phytochemicals and Plant-Based Whole Foods in Reducing LDL-C and Preventing Cardiovascular Disease. Antioxidants, 2021. 10(5).
2.Aslam, M.; Ladilov, Y. Emerging Role of cAMP/AMPK Signaling. Cells 2022, 11.
- In my opinion, the basic structures of reported other phytochemicals can be added in supplementary material, not in the manuscript.
Response: Thank you for your comments. We agree with your suggestion and agree to add Figure 4 to the supplementary material. Thank you again for your comments.
Fig. 4. Basic structures of other phytochemicals that have been found to have possible synergistic hypolipidemic effects.
Please see the attachment.
